# Characterization of two thermophilic cellulases from *Talaromyces leycettanus* JCM12802 and their synergistic action on cellulose hydrolysis

Yuan Gu[1☯], Fei Zheng[1,2☯], Yuan Wang[1], Xiaoyun Su[1], Yingguo Bai[1], Bin Yao[1], Huoqing Huang[1]*, Huiying Luo📷[1]*

**1** Key Laboratory for Feed Biotechnology of the Ministry of Agriculture, Feed Research Institute, Chinese Academy of Agricultural Sciences, Beijing, People's Republic of China, **2** College of Biological Sciences and Biotechnology, Beijing Forestry University, Beijing, People's Republic of China

☯ These authors contributed equally to this work.
* luohuiying@caas.cn (HL); huanghuoqing@caas.cn (HH)

**Data Availability Statement:** All relevant data are within the manuscript and its Supporting Information files.

## Abstract

*Talaromyces leycettanus* JCM12802 is a great producer of thermophilic glycoside hydrolases (GHs). In this study, two cellulases (*Tl*Cel5A and *Tl*Cel6A) belonging to GH5 and GH6 respectively were expressed in *Pichia pastoris* and functionally characterized. The enzymes had acidic and thermophilic properties, showing optimal activities at pH 3.5–4.5 and 75–80°C, and retained stable at temperatures up to 60°C and over a broad pH range of 2.0−8.0. *Tl*Cel5A and *Tl*Cel6A acted against several cellulose substrates with varied activities (3,101.1 vs. 92.9 U/mg to barley β-glucan, 3,905.6 U/mg vs. 109.0 U/mg to lichenan, and 840.3 and 0.09 U/mg to CMC-Na). When using Avicel, phosphoric acid swollen cellulose (PASC) or steam-exploded corn straw (SECS) as the substrate, combination of *Tl*Cel5A and *Tl*Cel6A showed significant synergistic action, releasing more reduced sugars (1.08–2.87 mM) than the individual enzymes. These two cellulases may represent potential enzyme additives for the efficient biomass conversion and bioethanol production.

## Introduction

Cellulose, the major component of plant cell walls, is composed of repeating D-glucose residues linked by β-1,4-glucosidic bonds. It represents the most plentiful sustainable resource on Earth and is widely used for the production of renewable energy [1,2,3]. The complete hydrolysis of cellulose requires the synergy of three types of cellulases: endoglucanase (EG; EC 3.2.1.4) that randomly cleaves the intracellular β-1,4-glucosidic bond of cellulose to release cellooligosaccharides; cellobiohydrolase (CBH; EC 3.2.1.91) that hydrolyzes short cellulose molecules and cellooligosaccharides to cellobiose; and β-glucosidase (BG; EC 3.2.1.21) to degrade cellobiose into glucose [4,5,6]

Based on the similarities of protein sequences and structures, cellulases are divided into different glycoside hydrolase (GH) families. EGs are often found in GH5, GH7, GH12 and GH45,

**Funding:** This research was supported by the National Natural Science Foundation of China (31572446), the China Modern Agriculture Research System (CARS-41), and the Collaborative Innovation Project of Chinese Academy of Agricultural Sciences, CAAS-463 XTCX2016011-04-5 (http://www.nsfc.gov.cn/english/site_1/) to HL. The funders had no role in study design, data collection and analysis, decision to publish, or preparation of the manuscript.

**Competing interests:** The authors have declared that no competing interests exist.

most CBHs belong to GH6 and GH7, and BGs are classified into GH1 and GH3 [7]. Among them, the EG of GH5 has been identified as the key endoglucanase of cellulase cocktail, and exhibits the highest catalytic efficiency in biomass conversion [8,9] and broad substrate specificity including cellulase, xylanase and mannanase activities [10,11,12]. The CBH of GH6 has capability of rapidly depolymerizing cellulose and cellooligosaccharides and is responsible for the majority of hydrolytic turnover [13], thus representing another key component of the cellulase cocktail [9].

Microbial cellulases have received great research and development attention, and are widely used in the biofuel, laundry, textile, food, animal feed, and pulp and paper industries [14,15]. Most of the commercial cellulases are derived from fungi, such as *Trichoderma*, *Humicola*, *Aspergillus* and *Penicillium*, due to the high activities and yields [16]. Dozens of GH5 EGs and GH6 CBHs have been cloned from fungi, expressed in prokyotic and eurakyotic systems, and functionally characterized [8,13,17,18]. Of them, those with thermostable properties are more favorable because of rapid viscosity reduction, lower process cost and fewer risks of microbial contamination [19–22]. Although most known fungal cellulases have optimal temperatures of 40–60˚C [9,23], some exceptions have thermophilic properties. For example, the Cel5H from *Dictyoglomus thermophilum* is active at 50–85˚C and retains thermostable at 70˚C [24]; the EGL2 from *Humicola grisea* is optimally active at 75˚C and remains more than 80% residual activity after incubation at 75˚C for 10 min, and the glucanase E2 from *Fusarium verticillioides* has a temperature optimum at 80˚C and keeps thermostable at 60˚C [25]. For fungal cellulases of GH6, only the CBHII from *Trichoderma viride* CICC13038 has thermophilic property with the optimal temperature of 70˚C [26].

*Talaromyces leycettanus* JCM12802 is a well-known producer of thermophilic GHs, including β-mannanase [27], β-glucosidase [28], polygalacturonases [19], xylanase [29], and α-galactosidase [23]. In the present study, the genes (*Tlcel5A* and *Tlcel6A*) coding for a GH5 endoglucanase and a GH6 cellobiohydrolase were identified in *T. leycettanus*. The gene products were then expressed in *Pichia pastoris*, biochemically characterized, and their synergistic action on the hydrolysis of Avicel, phosphoric acid swollen cellulose (PASC) and steam-exploded corn straw (SECS) at 60˚C were also analyzed [30].

## Materials and methods

### Strains, plasmids and culture conditions

*T. leycettanus* JCM12802 from the Japan Collection of Microorganisms (JCM, Japan) was used as the donor. It was cultivated at 40˚C in potato-dextrose broth (PDB) or complex medium containing 5 g/L $(NH_4)_2SO_4$, 1 g/L $KH_2PO_4$, 0.5 g/L $MgSO_4·7H_2O$, 0.2 g/L $CaCl_2$, 10 mg/L $FeSO_4·7H_2O$, 30 g/L corncob, 30 g/L soybean meal and 30 g/L wheat bran. *Escherichia coli* Trans1-T1 used as the cloning host was cultivated in liquid Luria-Bertani (LB) medium with ampicillin at 37˚C. The expression host *P. pastoris* GS115 (Invitrogen, Carlsbad, CA) was cultivated in the yeast peptone dextrose (YPD) medium at 30˚C. Minimal dextrose (MD) medium or minimal methanol (MM) medium, buffered glycerol complex (BMGY) medium, and buffered methanol complex (BMMY) medium were prepared according to the manual of the *Pichia* Expression kit (Invitrogen) and used for transformant selection, cultivation and induction, respectively. The vectors pGEM-T Easy (Promega, Madison, WI) and pPIC9 (Invitrogen) were used for cloning and expression, respectively.

### Reagents and chemicals

The restriction endonucleases, T4 DNA ligase and endo-β-*N*-acetylglucosaminidase H (Endo H) were purchased from New England Biolabs (Ipswich, MA). The *Taq* DNA polymerase was from

TaKaRa (Dalian, China). The DNA isolation and purification kits were purchased from Tiangen (Beijing, China) and Omega (Cowpens, SC), respectively. RNeasy plant mini kit and ReverTra-α-™ kit were supplied by Qiagen (QIAGEN, Germany) and TOYOBO (Osaka, Japan), respectively. The substrates lichenan, barley β-glucan, laminarin, Avicel, carboxymethyl cellulose sodium (CMC-Na), birchwood xylan, xyloglucan, konjac glucomannan, locus bean gum, 4-nitrophenyl β-D-cellobioside (*p*NPC), *p*-nitrophenyl-β-D-glucopyranoside (*p*NPGlu), and 4-nitrophenyl-α-D-galactopyranoside (*p*NPGal) were purchased from Sigma-Aldrich (St. Louis, MO). PASC treated by 85% of phosphoric acid and SECS were supplied by the Institute of Process, Chinese Academy of Sciences. And all other chemicals were of analytical grade and commercially available.

## Cloning of the cellulase-encoding genes

The full length cellulase-encoding genes, *Tlcel5A* and *Tlcel6A*, were identified in the genome sequence of *T. leycettanus* JCM12802, and obtained by PCR amplification with specific primers GH5F/GH5R and GH6F/GH6R (Table 1). After three days' growth in complex medium, the total RNA was extracted from the mycelia of *T. leycettanus* JCM12802 using the Qiagen RNEasy plant mini kit. cDNAs were then synthesized *in vitro* using the ReverTra Ace-a-™ kit. The cDNAs coding for the mature *Tl*Cel5A and *Tl*Cel6A were amplified with primers GH5F1/GH5R1 and GH6F1/GH6R1 harboring restriction sites, respectively. The PCR products with appropriate sizes were inserted into plasmid pGEM-T Easy and then transformed into *E. coli* Trans1-T1. Positive clones were confirmed by PCR and DNA sequencing.

## Sequence and structure analysis

The nucleotide and amino acid sequences were used for BLAST analysis at NCBI (http://www.ncbi.nlm.nih.gov/BLAST/). Simplified sequence alignment and prediction of the isoelectric point and molecular weight were conducted by using the Vector NTI Suite 10.0 software. Signal peptide prediction was carried out with SignalP 4.1 Server (http://www.cbs.dtu.dk/services/SignalP/). Multiple alignment of protein sequences was performed with the ClustalX 2.1 (http://www.clustal.org, [31], and the results were demonstrated by ESPript 3.0 (http://espript.ibcp.fr/ESPript/cgi-bin/ESPript.cgi [32]. The NetNglyc server was used to predict the putative *N*-glycosylation sites (http://www.cbs.dtu.dk/services/NetNGlyc/). And the modeled structures were predicted and visualized by using the SWISS-MODEL [33] and Pymol [34].

## Heterogeneous expression in *P. pastoris*

Heterologous expression of *Tl*Cel5A and *Tl*Cel6A was performed in *P. pastoris* according to the manual of the *Pichia* Expression kit (Invitrogen) with some modifications. The cDNA

**Table 1. Primers used in this study.**

| Primers | Sequences (5′→3′)[a] | Size (bp) |
| --- | --- | --- |
| GH5F | CCGTACGTAGCACCCAAGAGCAAGACCAAGC | 31 |
| GH5R | GATTGCGGCCGCCTACAGGCACTGGTAGTAATAGGGGTTC | 40 |
| GH6F | CATGAATTCCAGCAAACCATGTGGGGTCAATGC | 33 |
| GH6R | GATTGCGGCCGCCTAGAAAGAGGGGTTGGCGTTGGTAAG | 39 |
| GH5F1 | CCGTACGTAGCACCCAAGAGCAAGACCAAGC | 31 |
| GH5R1 | GATTGCGGCCGCCTACAGGCACTGGTAGTAATAGGGGTTC | 40 |
| GH6F1 | CATGAATTCCAGCAAACCATGTGGGGTCAATGC | 33 |
| GH6R1 | GATTGCGGCCGCCTAGAAAGAGGGGTTGGCGTTGGTAAG | 39 |

[a] The restriction sites are underlined

fragments coding for the mature proteins of *Tl*Cel5A and *Tl*Cel6A were digested with restriction enzymes *Sna*bI or *Eco*RI and *Not*I and subcloned into plasmid pPIC9 to yield expression vectors pPIC9-*Tlcel5A* and pPIC9-*Tlcel6A*, respectively. The recombinant plasmids were then linearized with *Bgl*II and individually transformed into *P. pastoris* GS115 competent cells by electroporation. Transformants were selected on MD plates, and positive transformants were grown in 10 mL shake tubes containing 3 mL BMGY for 2 days and 1 mL BMMY for another 2 days (220 rpm and 30˚C). The culture supernatants were collected by centrifugation at 5,000 rpm for 10 min, followed by cellulase activity assay as described below. The transformants with highest cellulase activities were selected for flask-level fermentation.

The most active transformants of *Tl*Cel5A and *Tl*Cel6A were grown in 1 L shake-flasks containing 400 mL BMGY at 220 rpm and 30˚C for 2 days, respectively. The cultures were pelleted by centrifugation and resuspended in 200 mL of BMMY for 3-day-growth at 30˚C and 200 rpm. Methanol was then added at the final concentration of 0.5% every 24 h for continuous induction of 3 days. The culture supernatants were collected by centrifugation at 4500× *g* for further analysis.

## Purification of recombinant *Tl*Cel5A and *Tl*Cel6A

The viva flow 200 ultrafiltration membrane system (Sartorius, Germany) with 5 kDa cut-off was used for the concentration and buffer exchange (to 0.1 M Tris-HCl, pH 8.0) of the crude enzymes. The recombinant proteins were desalted by HiTrapTM Desalting column and purified using the HiTrap Q Sepharose XL FPLC column (GE Healthcare) pre-equilibrated with 0.1 M Tris-HCl (pH 8.0). The gradient NaCl of 0–0.7 M at the flow rate of 3 mL/min was used to elute the proteins. Fractions showing cellulase activities were pooled and further desalted with a 5 kDa molecular cut-off concentration tube (Millipore) using 0.1 M McIlvaine buffer (pH 3.5 or 4.5). The purified proteins were separated on 12% sodium dodecyl sulfate-polyacrylamide gel electrophoresis (SDS-PAGE) for purity and molecular mass analysis. Protein concentration was determined by using the Bradford method.

## Deglycosylation of the purified recombinant *Tl*Cel5A and *Tl*Cel6A

To remove *N*-glycosylation, the purified recombinant *Tl*Cel5A and *Tl*Cel6A were treated with Endo H for 2 h at 37˚C following the manufacturer's instructions (New England Biolabs), and checked by SDS-PAGE.

## Enzyme activity assays

The cellulase activities were determined using 1% (w/v) CMC-Na, barley β-glucan or lichenan as the substrate. The reaction mixtures containing 100 μL of properly diluted enzyme solution and 900 μL substrate solution in 0.1 M McIlvaine buffer (pH 3.5 and 4.5) were incubated at 75˚C or 80˚C (optimum temperature) for 10 min. When using 5 mg/mL Avicel as the substrate, the reaction period was lengthened to 60 min. The amounts of reducing sugars were determined via the 3,5-dinitrosalicylic acid (DNS) method [35]. One unit (U) of enzyme activity was defined as the amount of enzyme producing 1 μmol of reducing sugar per minute under the assay conditions.

## Biochemical characterization of *Tl*Cel5A and *Tl*Cel6A

CMC-Na, barley β-glucan and PASC were used as the substrates for enzyme characterization. The optimal pH was determined at 75˚C for *Tl*Cel5A and 80˚C for *Tl*Cel6A, respectively, in 0.1 M McIlvaine buffer with pH ranging from 2.0 to 8.0. To test the pH stability, the purified

enzymes were pre-incubated at 37˚C for 60 min in buffers with pH ranging from 1.0 to 12.0 (0.1 M KCl-HCl for pH 1.0–2.0, 0.1 M McIlvaine buffer for 2.0 to 8.0, and 0.1 M glycine-NaOH for pH 9.0–12.0). The residual enzyme activities were determined at 75˚C and pH 3.5 for *Tl*Cel5A and 80˚C (barley β-glucan) or 70˚C (PASC) and pH 4.5 for *Tl*Cel6A, respectively. To determine the optimal temperature, the cellulase activities were determined at temperatures ranging from 30˚C to 90˚C and pH 3.5 for *Tl*Cel5A or pH 4.5 for *Tl*Cel6A for 10 min, respectively. For thermal stability assays, *Tl*Cel5A and *Tl*Cel6A were incubated at temperatures of 70˚C, 75˚C or 80˚C and pH 3.5 or 4.5 for different periods of time. For half-life determination, 0.05 mg of *Tl*Cel5A or 0.2 mg of *Tl*Cel6A was incubated at 60˚C, 65˚C or 70˚C for 0.5–24 h. Residual activities were determined as described above.

## Substrate specifiity and kinetic parameters

Substrate specificity of *Tl*Cel5A and *Tl*Cel6A were determined by using 1% (w/v) CMC-Na, barley β-glucan, Avicel, lichenan, laminarin, xyloglucan, birchwood xylan, konjac flour, locus bean gum, glucomannan, or 0.2 mM of *p*NPGlu, *p*NPGal, *p*NPC as the substrate.

The kinetic parameters ($K_m$, $V_{max}$ and $k_{cat}/K_m$) of *Tl*Cel5A were determined by incubating the enzyme with 0.25 to 10 mg/mL CMC-Na at pH 3.5 and 75˚C for 5 min. For *Tl*Cel6A, the kinetic parameters were determined at pH 4.5 and 80˚C for 5 min with 0.25–10.0 mg/mL barley β-glucan as the substrate. The enzyme activities were determined by using the DNS method. The kinetic constants were calculated using the Lineweaver-Burk plots by GraphPad Prism 6.0 (http://www.graphpad.com/scientific-software/prism/).

## Analysis of the cellooligosaccharides hydrolysis products

The hydrolysis products of cellooligosaccharides by *Tl*Cel5A and *Tl*Cel6A were detected by high-performance anion exchange chromatography (HPAEC, model 2500, Dionex, Sunnyvale, CA). Purified *Tl*Cel5A (0.05 U) or *Tl*Cel6A (0.02 U) was added into 1 mL of cellooligosaccharide solution containing 200 μg of cellotetraose, cellopentaose, or cellohexaose and incubated at 60˚C for 0 min, 1 min, 5 min, 30 min, 1 h, 4 h or 5 h. After enzyme inactivation by boiling water bath, the hydrolysates were diluted 100 times with ddH$_2$O, and 100 μL of each sample was injected into the column of HPAEC. The oligosaccharides were eluted by 100 mM NaOH. The standards consisted of glucose, cellobiose, cellotriose, cellotetraose, cellopentaose, and cellohexaose. The amount of each hydrolysate was calculated based on the peak area. One enzyme unit (U) was defined as the amount of enzyme to hydrolyze 1 μmol of substrate per minute under the assay conditions. The catalytic efficiency values ($k_{cat}/K_m$) of *Tl*Cel6A against cellooligosaccharides were calculated according to Xia et al. (2016).

## Synergistic action of *Tl*Cel5A and *Tl*Cel6A on cellulose hydrolysis

To determine the hydrolytic capabilities of *Tl*Cel5A and *Tl*Cel6A to degrade cellulose substrates, the enzymes were combined at different ratios, and the commercial β-glucosidase from *Aspergillus fumigatu* (Sigma-Aldrich) was added at the concentration of 10% (mol β-glucosidase/mol total enzyme) to convert cellobiose to glucose. When using Avicel as the substrate, *Tl*Cel5A and *Tl*Cel6A were combined at the ratios of 0.4:3.6, 0.8:3.2, 1.2:2.8, 1.6:2.4, 2.0:2.0, 2.4:1.6, 2.8:1.2, 3.2:0.8, 3.6:0.4 in the total amount of 4.0 μM and incubated with 5 mg/mL of Avicel in 50 mM McIlvaine buffer, pH 4.0 at 60˚C for 24 h. The amounts of reducing sugars released were determined using the DNS method.

Based on the results above, the best ratio of *Tl*Cel5A and *Tl*Cel6A was used for further enzyme combinations. When using SECS or Avicel as the substrate, 8 μM of *Tl*Cel5A or *Tl*Cel6A alone, or 4 μM *Tl*Cel5A and 4 μM *Tl*Cel6A was incubated with 5 mg/mL substrate at

pH 4.0 and 60°C for 12 or 18 h. With 2.5 mg/mL PASC as the substrate, 2 μM of *Tl*Cel5A or *Tl*Cel6A alone or 1 μM *Tl*Cel5A and 1 μM *Tl*Cel6A was added. The amounts of reducing sugars released were determined by using the DNS method. The amounts of glucose released were analyzed by a one-factor ANOVA of SPSS19.0 to assess the synergistic effects of *Tl*Cel5A and *Tl*Cel6A. Statistical differences were considered to be significant at $P < 0.05$.

### Nucleotide sequence accession numbers

The nucleotide sequences of *Tl*Cel5A and *Tl*Cel6A from *T. leycettanus* JCM12802 have been deposited in the GenBank database under the accession numbers MG993208 and MG993209, respectively.

### Labooratory protocol

Link: dx.doi.org/10.17504/protocols.io.8g4htyw

## Results

### Gene cloning and sequence analysis

The full-length *Tlcel5A* and *Tlcel6A* contained 1464 bp and 1898 bp, respectively, which were interrupted by four (67, 56, 57, and 54 bp) and seven (69, 75, 78, 69, 71, 70, and 71 bp) introns. The cDNAs were 1230 bp and 1398 bp in lengths, and encoded two polypeptides of 409 and 465 amino acids, respectively. A putative signal peptide of 18 amino acids was predicted at the N-termini, and two (Asn36 and Asn240 in *Tl*Cel5A) and three (Asn293, Asn307 and Asn423 in *Tl*Cel6A) potential *N*-glycosylation sites were predicted in deduced *Tl*Cel5A and *Tl*Cel6A.

BLAST analysis indicated that *Tl*Cel5A is an endoglucanase of GH5 while *Tl*Cel6A is a cellobiohydrolase of GH6, respectively. Both enzymes consist of a cellulose-binding module (CBM), a linker region and a catalytic domain. The CBM of *Tl*Cel5A is at the C-terminus while that of *Tl*Cel6A is at the N-terminus. The deduced *Tl*Cel5A showed the highest protein identity of 76% to an EG from *Aspergillus udagawae* (GAO86105.1) and 77% with the catalytic domain of structure-resolved EG (1GZJ) from *Thermoascus aurantiacus*. Deduced *Tl*Cel6A shared the highest identity of 99% with a GH6 CBH from *T. leycettanus* (CDF76448.1).

Multiple sequence alignment and structure analysis (S1 Fig) indicated that Glu159 and Glu266 of *Tl*Cel5A and Asp239 and Asp419 of *Tl*Cel6A function as the proton donors and nucleophiles, respectively. *Tl*Cel5A has eight conserved residues of GH5 members, including Arg75, His119, Asn158, Glu159, His224, Tyr226, Glu266 and Trp299. The catalytic domain of *Tl*Cel5A was a typical $(\alpha/\beta)_8$ barrel structure (S2A Fig) containing a long and shallow substrate binding cavity. Several aromatic residues in the cavity played very important role in enzyme activity. Thirteen conserved residues (Tyr120, Trp151, Tyr187, Asp193, Asp239, Asn243, Thr246, Trp287, Trp290, Trp382, Trp385, Asp419, and His432) of GH6 members were found in *Tl*Cel6A. The modeled structure of the catalytic domain of *Tl*Cel6A is a distorted α/β barrel (S2B Fig), in which the central β-sheet core is composed of seven instead of eight parallel β-strands. It has a long substrate binding groove, and the catalytic center is enclosed by two active-site loops (residues 190–207 and 412–446). The loops are stabilized by disulfide bonds Cys194-Cys253 and Cys386-Cys433.

### Expression and purification of the recombinant *Tl*Cel5A and *Tl*Cel6A

Recombinant *Tl*Cel5A and *Tl*Cel6A were successfully expressed in *P. pastoris* GS115. With barley β-glucan as the substrate, the highest activities of 2.02 U/mL and 1.86 U/mL, were detected in the positive transformants of *Tl*Cel5A and *Tl*Cel6A, respectively. After large-scale

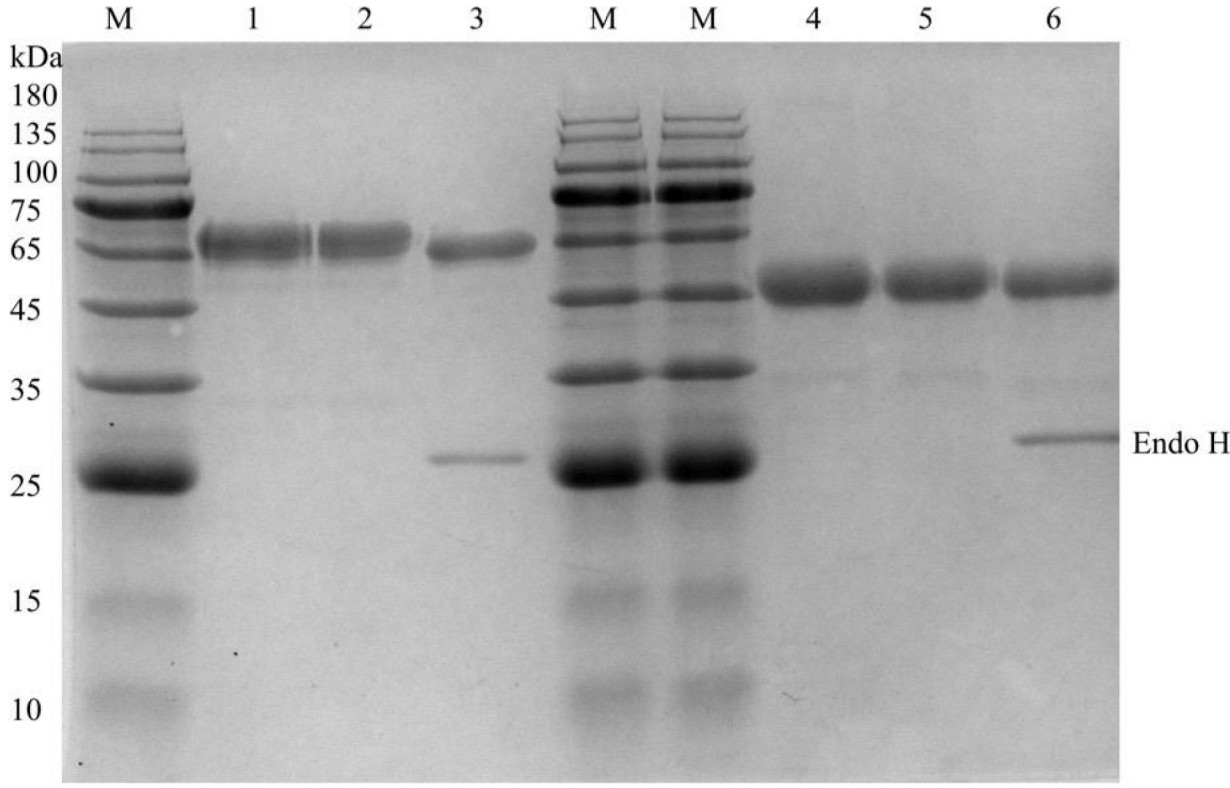

**Fig 1. SDS-PAGE (12% polyacrylamide gel) analysis of the purified recombinant *Tl*Cel5A and *Tl*Cel6A.** Lane M, the molecular mass standards; lane 1, the culture supernatant of *Tl*Cel6A; lane 2, the purified recombinant *Tl*Cel6A; lane 3, the deglycosylated *Tl*Cel6A with Endo H treatment; lane 4, the culture supernatant of *Tl*Cel5A; lane 5, the purified recombinant *Tl*Cel5A; lane 6, the deglycosylated *Tl*Cel5A with Endo H treatment.

fermentation for 144 h, the culture supernatants were collected, concentrated and purified to electrophoretic homogeneity by anion exchange chromatography. SDS-PAGE revealed that the purified recombinant *Tl*Cel5A and *Tl*Cel6A had apparent molecular masses of approximately 45 kDa and 65 kDa (Fig 1), which were higher than their theoretical values (42.5 kDa and 47.0 kDa). After Endo H treatment, the molecular mass of *Tl*Cel5A remained the same, while the *Tl*Cel6A migrated as a single band of about 60 kDa. Other post-translational modification like *O*-glycosylation might account for the extra molecular masses.

### Enzymatic properties of purified recombinant *Tl*Cel5A and *Tl*Cel6A

The enzymatic properties of *Tl*Cel5A were determined with CMC-Na, and the *Tl*Cel6A activities were determined by using barley β-glucan and PASC as the substrates. *Tl*Cel5A had the optimal pH of 3.5 (Fig 2A) and retained more than 90% activity at pH 2.0 to 10.0 (Fig 2B). The pH optima of *Tl*Cel6A towards barley β-glucan and PASC were both 4.5 (Fig 2A), but retained stable over a narrower pH range of 2.0 to 8.0 (Fig 2B). Both enzymes were thermophilic: *Tl*Cel5A was optimally active at 75°C, and remained more than 90% activity at 80°C, while *Tl*Cel6A showed the maximum activity at 80°C to barley β-glucan and 70°C to PASC (Fig 2A). The enzymes were highly stable at 60°C (Fig 2D). When increased the temperature to 70°C, *Tl*Cel5A retained most of the activity after 1-h incubation, while *Tl*Cel6A lost approximately 50% and 40% activities towards barley β-glucan and PASC, respectively, after 30 min (Fig 2E

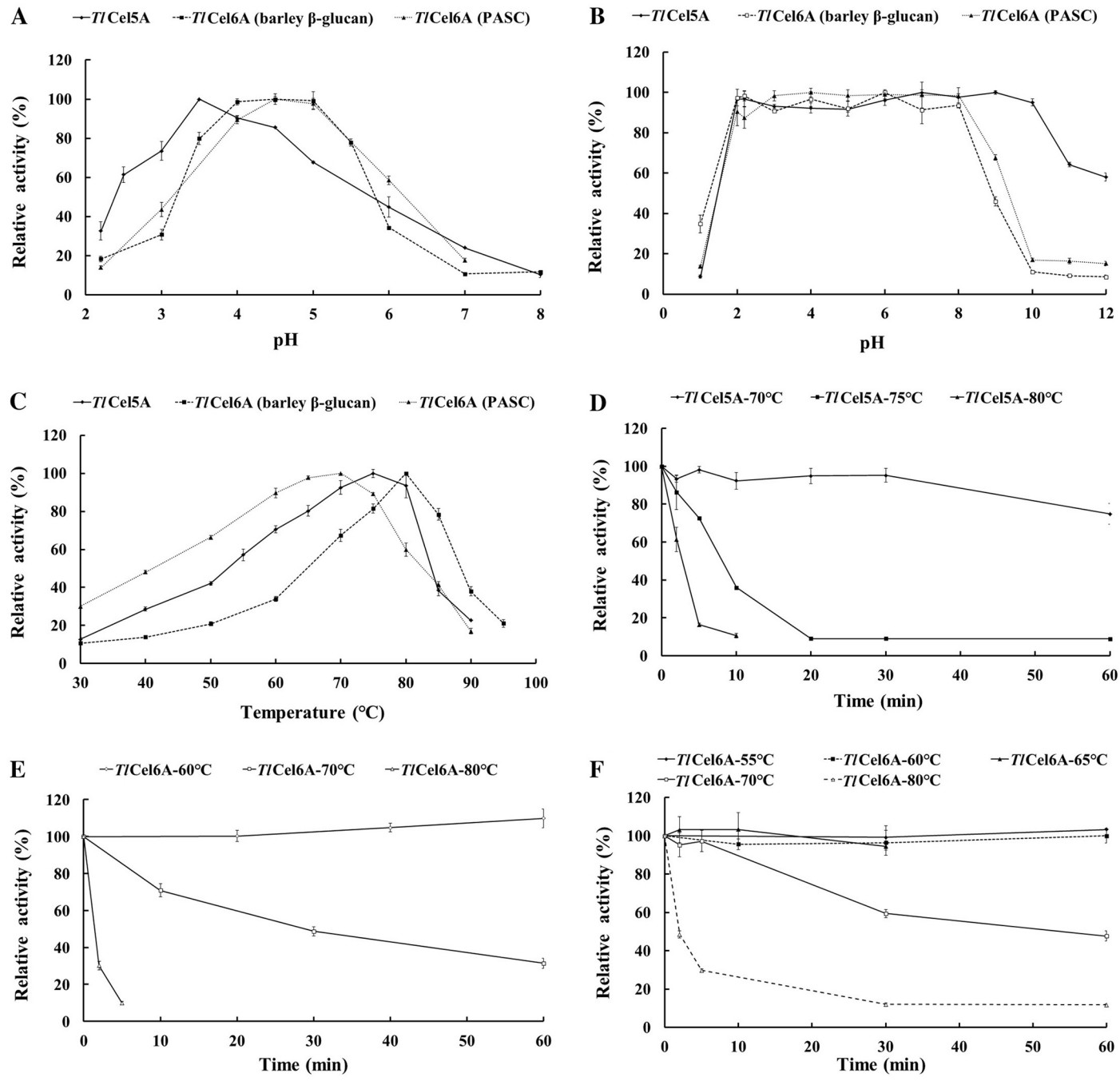

**Fig 2. Effects of pH and temperature on the enzyme activity and stability of purified recombinant *Tl*Cel5A and *Tl*Cel6A.** (A) pH-activity profiles. *Tl*Cel5A was incubated at 75°C for 10 min with 1.0% (w/v) CMC for enzyme assays. The enzymatic activities of *Tl*Cel6A were measured at 70°C against barley β-glucan and 80°C for PASC for 10 min, respectively. (B) pH stability of *Tl*Cel5A and *Tl*Cel6A after 1 h-incubation at 37°C. The enzyme activities were measured under standard conditions. (C) Temperature-activity profiles of *Tl*Cel5A and *Tl*Cel6A assayed at optimal pH for 10 min. (D) *Tl*Cel5A thermostability at 70, 75 and 80°C and pH 3.5 for various durations. (E) *Tl*Cel6A thermostability at 60, 70 and 80°C and pH 4.5 with barley β-glucan as the substrate. (F) *Tl*Cel6A activities measured at 55, 60, 65, 70 and 80°C and pH 4.5 with PASC as the substrate. Each value in the panel represents the means ± SD (n = 3).

and 2F). The $t_{1/2}$ values of $Tl$Cel5A and $Tl$Cel6A were determined to be 20 h and 31 h at 60°C, 4.2 h and 7 h at 65°C, and 1.7 h and 0.5 h at 70°C.

## Substrate specificity and kinetic parameters

$Tl$Cel5A exhibited broad substrate specificity including barley β-glucan, lichenan, CMC-Na, Avicel, laminarin, konjac glucomannan, and birchwood xylan. It had higher specific activities on lichenan (3,905.6 ± 39.9 U/mg), barley β-glucan (3,101.1 ± 12.1 U/mg) than on CMC-Na (840.3 ± 0.7 U/mg). With CMC-Na as the substrate, the $K_m$, $V_{max}$ and $k_{cat}/K_m$ values of purified $Tl$Cel5A were determined to be 7.06 mg/mL, 1130.2 μmol/min/mg, and 113.4 mL/s/mg, respectively.

The specific activities of $Tl$Cel6A on lichenan, barley β-glucan and PASC were 109.0 ± 4.5 U/mg, 92.9 ± 1.4 U/mg and 4.0 ± 0.1 U/mg. When using Avicel and CMC-Na as the substrate, $Tl$Cel6A showed much lower activities (0.12 U/mg and 0.09 U/mg). No activity was detected on $p$NP-substrates. The $K_m$, $V_{max}$, and $k_{cat}/K_m$ values of $Tl$Cel6A using barley β-glucan as the substrate were determined to be 2.1 mg/mL, 196.1 μmol/min/mg, and 73.2 mL/s/mg, respectively. For cellooligosaccharide hydrolysis, $Tl$Cel5A had low specific activity on cellotetraose, cellopentaose (15.2 U/mg) and cellohexaose (16.1 U/mg). And $Tl$Cel6A had high activities on cellotetraose (186.7 U/mg), cellopentaose (99.6 U/mg), and cellohexaose (138.9 U/mg). The relative low activity on cellopentaose might be ascribed to the preference of $Tl$Cel6A for cellooligosaccharides containing even numbers of glucose units.

## Hydrolysis products of cellooligosaccharides

The hydrolysis products of $Tl$Cel5A and $Tl$Cel6A on cellooligosaccharides (cellotetraose, cellopentaose, or cellohexaose) were analyzed by using the method of HPLC (Table 2). $Tl$Cel5A had no activity on cellotetraose (S3A Fig), cleaved cellopentaose into mainly cellobiose and cellotriose and low amounts of cellotetraose (S3B Fig), and cellohexaose into cellobiose, cellotriose, cellotetraose and cellopentaose (S3C Fig). $Tl$Cel6A was active against cellotetraose, releasing cellobiose as the sole product (S3D Fig). $Tl$Cel6A also hydrolyzed cellopentaose and cellohexaose rapidly. The main products were cellobiose and cellotriose (S3E and S3F Fig). The catalytic efficiency values of $Tl$Cel5A and $Tl$Cel6A against cellotetraose, cellopentaose and cellohexaose were 0, 10.4, 19.3 and 95.2, 42.9, 139. 3 /μM/min, respectively.

## Synergistic action of $Tl$Cel5A and $Tl$Cel6A on cellulose hydrolysis

The hydrolytic capabilities of $Tl$Cel5A, $Tl$Cel6A and their combinations at different concentrations (0.4–4.0 μM) towards Avicel were determined after 24-h incubation at 60°C. As shown in Fig 3A, the enzyme combinations released more glucose than the individual enzyme or the sum of the individuals. The highest amount of glucose (4.00 mM) was released by the enzyme combination of 2 μM $Tl$Cel5A and 2 μM $Tl$Cel6A. Thus the enzyme ratio of 1:1 was used for further studies.

The combination of $Tl$Cel5A and $Tl$Cel6A released 3.72 mM, 4.33 mM, and 7.87 mM of glucose from Avicel at 18 h, SECS at 12 h, and PASC at 60 min, respectively, which were significantly higher ($P < 0.05$) than that released by $Tl$Cel5A or $Tl$Cel6A individually from Avicel (1.08 and 2.64 mM; Fig 3B), SECS (1.46 and 1.96 mM; Fig 3C), and PASC (5.33 and 5.51 mM; Fig 3D). The results indicated that $Tl$Cel5A and $Tl$Cel6A have synergistic action on cellulose degradation.

**Table 2. Hydrolysis product analysis of cellooligosaccharides by *Tl*Cel5A and *Tl*Cel6A.**

| Protein | Substrate | Time (min) | Oligosaccharides (µg) | | | | |
|---|---|---|---|---|---|---|---|
| | | | Cellobiose | Cellotriose | Cellotetraose | Cellopentaose | Cellohexaose |
| *Tl*Cel5A | Cellotetraose | 5 | - | - | 4.84±0.06 | | |
| | | 60 | - | - | 4.82±0.02 | | |
| | | 120 | - | - | 4.75±0.12 | | |
| *Tl*Cel5A | Cellopentaose | 5 | 0.34±0.01 | 0.41±0.01 | 0.34±0.02 | 5.76±0.03 | |
| | | 30 | 0.95±0.01 | 1.18±0.01 | 0.38±0.01 | 4.73±0.03 | |
| | | 120 | 2.09±0.01 | 2.57±0.02 | 0.45±0.01 | 2.94±0.01 | |
| | | 300 | 3.27±0.03 | 3.79±0.04 | 0.55±0.01 | 1.20±0.01 | |
| *Tl*Cel5A | Cellohexaose | 5 | 0.29±0.01 | 0.42±0.01 | 0.46±0.01 | - | 3.76±0.01 |
| | | 30 | 0.66±0.01 | 1.22±0.02 | 1.07±0.02 | - | 1.68±0.01 |
| | | 60 | 0.78±0.01 | 1.63±0,02 | 1.27±0.01 | - | 1.01±0.01 |
| | | 120 | 0.93±0.01 | 2.28±0.01 | 1.38±0.01 | - | 0.35±0.01 |
| *Tl*Cel6A | Cellotetraose | 1 | 3.43±0.05 | - | 2.65±0.06 | | |
| | | 5 | 6.53±0.06 | - | 0.21±0.01 | | |
| *Tl*Cel6A | Cellopentaose | 1 | 1.05±0.03 | 1.50±0.03 | - | 4.72±0.07 | |
| | | 5 | 2.10±0.03 | 3.00±0.03 | - | 3.05±0.02 | |
| | | 30 | 4.21±0.02 | 5.79±0.03 | - | - | |
| | | 60 | 4.22±0.02 | 5.79±0.02 | - | - | |
| *Tl*Cel6A | Cellohexaose | 1 | 1.09±0.02 | 1.56±0.02 | 0.51±0.01 | - | 1.81±0.03 |
| | | 5 | 2.15±0.01 | 2.92±0.01 | 0.50±0.01 | - | - |
| | | 30 | 2.24±0.01 | 2.74±0.02 | 0.31±0.01 | - | - |
| | | 60 | 2.38±0.01 | 2.76±0.01 | 0.20±0.01 | - | - |

## Discussion

The endoglucanases of GH5 and cellobiohydrolases of GH6 are crucial components of the cellulase complex for biomass conversion. These cellulases generally have a temperature optimum of 40–70˚C and retain stable at 50–60˚C [9,36,37]. To improve the hydrolytic efficiency and reduce the cost, thermostable cellulases with high activities are much favorable. In the present study, two cellulases (*Tl*Cel5A and *Tl*Cel6A) from *T. leycettanus* JCM12802 were heterologously produced in *P. pastoris* and functionally characterized. The recombinant enzymes were found to be thermophilic and highly active, and showed great efficiency in the bioconversion of various cellulose substrates.

In comparison to their close homologues, *Tl*Cel5A and *Tl*Cel6A showed adaptability to higher temperatures. For example, the optimal temperature of *Tl*Cel5A is 75˚C, which is at least 5˚C higher than those characterized EGs (with 60–77% sequence identities) of *Penicillium brasilianum* (Cel5C; [38], *Aspergillus niger* (*An*Cel5A; [39], and *Thermoascus aurantiacus* (EGI; [17]. One exception is the *Te*Egl5A from *T. emersonii* CBS394.64, which exhibits optimal activity at 90˚C [12]. However, the *Tl*Cel5A activity (840 U/mg) on CMC-Na was much higher than that of *Te*Egl5A (624 U/mg), EGI (336 U/mg) and Cel5C (51 U/mg). The temperature optima of *Tl*Cel6A on barley β-glucan and PASC are 80˚C and 70˚C, much higher than or similar to its homologues 40–70˚C [26,40,41]. These excellent properties make *Tl*Cel5A and *Tl*Cel6A much potential for the application in the biofuel industry.

The hydrolytic capabilities of *Tl*Cel5A and *Tl*Cel6A in combination with a commercial glucosidase were then assessed. These enzymes employ endo- and exo-mode of actions [42–44], and acted synergistically on the hydrolysis of different cellulose substrates. When added the *Tl*Cel5A and *Tl*Cel6A at the ratio of 1:1, it produced the highest amount of glucose (up to 4

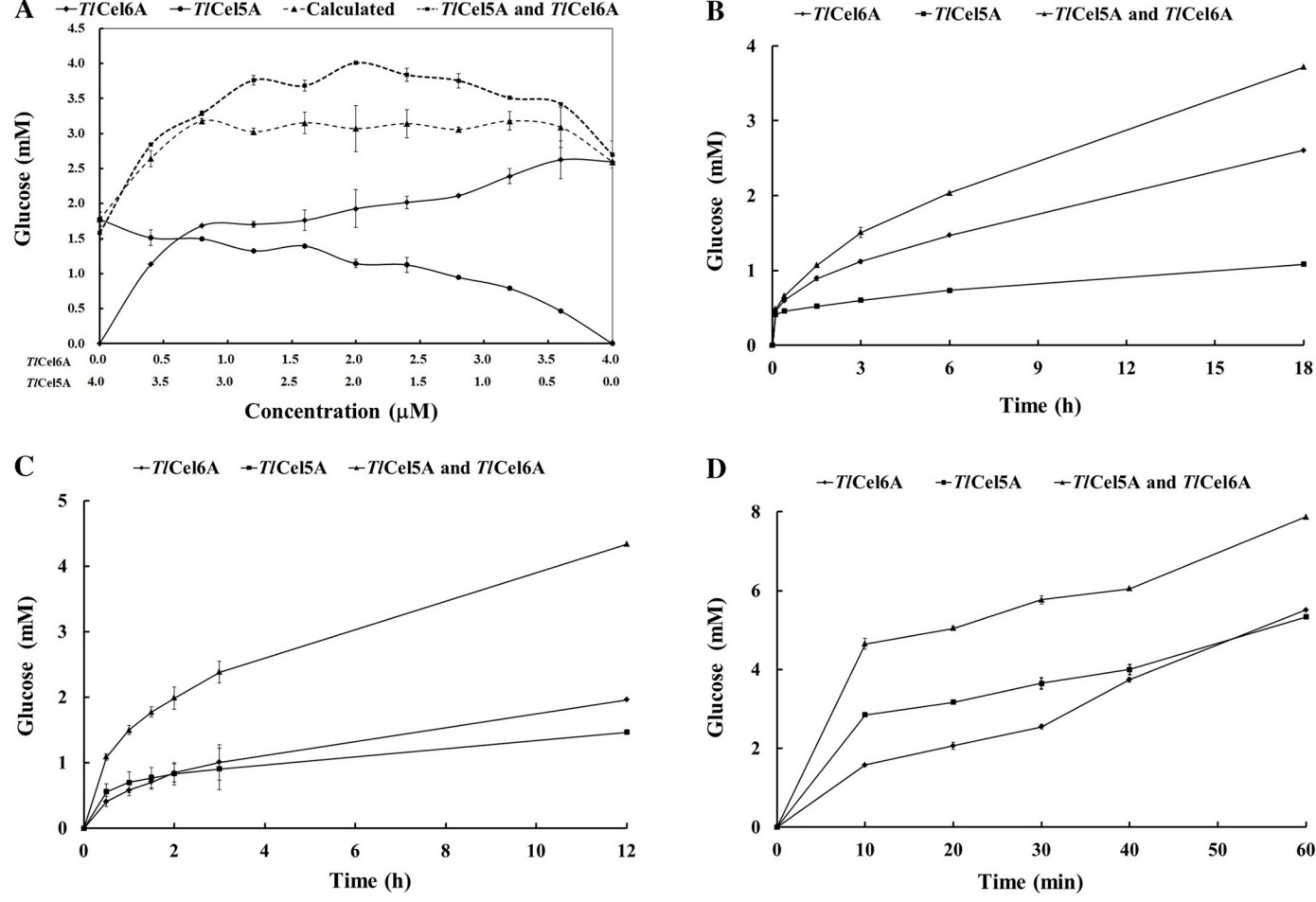

**Fig 3. Synergistic actions of *Tl*Cel5A, *Tl*Cel6A and a commercial glucosidase on the hydrolysis of different cellulose substrates.** (A)Avicel (5.0 mg/mL) hydrolysis by the *Tl*Cel5A, *Tl*Cel6A, and their combination at different ratios. (B)Avicel (2.5 mg/mL) hydrolysis by the *Tl*Cel5A, *Tl*Cel6A, and their combination at the ratio of 1:1. (C) SECS (5.0 mg/mL) hydrolysis by the *Tl*Cel5A, *Tl*Cel6A, and their combination at the ratio of 1:1. (D) PASC (2.5 mg/mL) hydrolysis by the *Tl*Cel5A, *Tl*Cel6A, and their combination at the ratio of 1:1. Each experiment had triplicate, and the data were shown as mean ± standard deviation (n = 3).

mM). There are significant differences in the amounts of reducing sugars released by *Tl*Cel5A, *Tl*Cel6A and their combination. The synergy effect of combined *Tl*Cel5A and *Tl*Cel6A with Avicel as the substrate is greater than that of *Ct*endo45 and *Ct*Cel6 [19], which might be ascribed to the enzymatic properties of different enzyme components. Although *Tl*Cel5A and *Tl*Cel6A individually showed higher activities on PASC than on Avicel, the synergy effect of *Tl*Cel5A and *Tl*Cel6A was lower on PASC. Similar results have been reported by Zhang and Lynd [45]. For natural substrate like SECS, *Tl*Cel5A and *Tl*Cel6A showed greater synergy than the individual enzymes. Of the two enzymes, *Tl*Cel6A had higher hydrolysis activity than *Tl*Cel5A towards crystalline Avicel, but had no difference in hydrolysis of PASC and steam-exploded corn straw. The reason might be that *Tl*Cel5A contributes to the increased hydrolytic efficiency of *Tl*Cel6A by producing new chain ends and smoothing the physical obstacles on the surface of cellulose, while *Tl*Cel6A depolymerizes some tight cellulose structure to promote the *Tl*Cel5A activity [46]. Therefore, the enzyme combination of *Tl*Cel5A and *Tl*Cel6A is more favorable to degrade natural biomass with higher hydrolysis efficiency.

It has been reported that *N*-linked glycan is one important component of the processive machinery of cellobiohydrolases and plays a role in the enzyme activity [47]. Although some studies indicated that *N*-glycosylation has no significant effect on enzyme activity, and it does influence the protein stability [48]. According to the theory of [47], the position of *N*-glycosylation on the protein surface plays the key role. There are three potential *N*-glycosylation sites (Asn293, Asn307 and Asn423) in *Tl*Cel6A. Since the glycan linked to Asn293 is located in close proximity to the entrance of the active site tunnel, this long glycan chain might block the cellulose molecules from the tunnel and finally decrease the activity. On the other hand, Asn423 is located on the active loop, and the glycans attached to this residue might form interactions with the cellulose molecules and boost the enzyme activity. Further site-directed mutagenesis at these potential *N*-glycosylation sites will be conducted to reveal the underlying mechanisms.

## Conclusions

Two acidic cellulases *Tl*Cel5A and *Tl*Cel6A were identified in *T. leycettanus* JCM12802 and heterologously produced in *P. pastoris*. In comparison to close homologues, these cellulases are superior in higher activities and greater thermostability. Their combination at the ratio of 1:1 showed distinguished synergy on Avicel, PASC and steam-exploded corn straw. Thus *Tl*Cel5A and *Tl*Cel6A may represent great candidates for the industrial conversion of lignocellulose.

## Supporting information

**S1 Fig. Multiple sequence alignment of *Tl*Cel5A (A) and *Tl*Cel6A (B) with known sequences.** The catalytic residues are indicated by circles. The conserved residues are indicated with asterisks. And the potential N-glycosylation sites are indicated by triangles.
(TIF)

**S2 Fig. Modeled structures of *Tl*Cel5A (A) and *Tl*Cel6A (B).** Orange sticks represent potential N-glycosylation sites, red sticks indicate the catalytic residues, and spheres indicate the conserved residues.
(TIF)

**S3 Fig. Hydrolysis products of *Tl*Cel5A and *Tl*Cel6A on cellooligosaccharides determined by HPLC.**
(TIF)

**S1 Raw Images.**
(TIF)

## Author Contributions

**Conceptualization:** Bin Yao, Huoqing Huang.

**Data curation:** Fei Zheng, Huiying Luo.

**Formal analysis:** Xiaoyun Su.

**Investigation:** Yingguo Bai.

**Methodology:** Huoqing Huang, Huiying Luo.

**Resources:** Yuan Wang.

**Supervision:** Bin Yao, Huoqing Huang, Huiying Luo.

**Writing – original draft:** Yuan Gu.

**Writing – review & editing:** Xiaoyun Su.

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
