## [Decision Letter · Decision Letter 0]

9 Sep 2019

PONE-D-19-21437

Characterization of two thermophilic cellulases from Talaromyces leycettanus JCM12802 and their synergistic action on cellulose hydrolysis

PLOS ONE

Dear Mrs. Luo,

Thank you for submitting your manuscript to PLOS ONE. After careful consideration, we feel that it has merit but does not fully meet PLOS ONE’s publication criteria as it currently stands. Therefore, we invite you to submit a revised version of the manuscript that addresses the points raised during the review process.

We would appreciate receiving your revised manuscript by Oct 24 2019 11:59PM. To enhance the reproducibility of your results, we recommend that if applicable you deposit your laboratory protocols in protocols.io, where a protocol can be assigned its own identifier (DOI) such that it can be cited independently in the future. For instructions see: http://journals.plos.org/plosone/s/submission-guidelines#loc-laboratory-protocols

We look forward to receiving your revised manuscript.

Kind regards,

Jean-Guy Berrin

Academic Editor

PLOS ONE

Journal Requirements:

Reviewers' comments:

Reviewer's Responses to Questions

**Comments to the Author**

1. Is the manuscript technically sound, and do the data support the conclusions?

Reviewer #1: Yes

Reviewer #2: Yes

2. Has the statistical analysis been performed appropriately and rigorously? 

Reviewer #1: Yes

Reviewer #2: Yes

3. Have the authors made all data underlying the findings in their manuscript fully available?

Reviewer #1: Yes

Reviewer #2: Yes

4. Is the manuscript presented in an intelligible fashion and written in standard English?

Reviewer #1: Yes

Reviewer #2: Yes

5. Review Comments to the Author

Reviewer #1: Figures are of poor quality and need to be replaced

Several of the references are old and should be replaced or supplemented with more recent publications

Reviewer #2: I recommend this manuscript for publication in PLOS ONE. I suggest the following issue to address though. Speaking about the synergistic action, the Authors should make some thoughts and provide chemical scheme of hydrolysis reaction pathway for this scenario. It will simplify understanding the concept of enzymes action.

6. PLOS authors have the option to publish the peer review history of their article (what does this mean?). If published, this will include your full peer review and any attached files.

Reviewer #1: No

Reviewer #2: No

---

## [Author Response · Author response to Decision Letter 0]

19 Oct 2019

Response to editor:

Answer: We have checked the manuscript carefully and changed the name of my files to ensure meet PLOS ONE's style requirements.

2. PLOS ONE now requires that authors provide the original uncropped and unadjusted images underlying all blot or gel results reported in a submission’s figures or Supporting Information files.

 Answer: I have provided the original uncropped and unadjusted gel image in the Supporting Information file.

3. In your cover letter, please note whether your blot/gel image data are in Supporting Information or posted at a public data repository.

 Answer: The gel image data has been provided in the Supporting Information file.

4. Figures are of poor quality and need to be replaced.

 Answer: The figures with high quality have been replaced and uploaded.

5. To enhance the reproducibility of your results, we recommend that if applicable you deposit your laboratory protocols in protocols.io, where a protocol can be assigned its own identifier (DOI) such that it can be cited independently in the future.

Answer: I have finished the protocol and uploaded it, this is the link: dx.doi.org/10.17504/protocols.io.8g4htyw.

Response to Reviewer 1:

Reviewer #1: Several of the references are old and should be replaced or supplemented with more recent publications

Answer: The latest references have been added and inserted into the corresponding position in the article and marked yellow. The followings are the six newly inserted references.

3. Touijer H, Benchemsi N, Ettayebi M, Janati A, Chaouni B, Bekkari H. Thermostable Cellulases from the Yeast Trichosporon sp. Enzyme research. 2019;1-6. (Line42) 

6. Kamal S, Khan SU, Muhammad N, Shoaib M, Omar M, Kaneza P, et al. Insights on heterologous expression of fungal cellulases in Pichia pastoris. Biochemistry and Molecular Biology. 2018; 3(1): 15-35. (Line47)

15. Kamal S, Khan SU, Khan S, Shoaib M, Khan H, Man S, et al. Recent view on heterologous expression of thermostable fungal cellulases, focused on expression factory of Pichia Pastoris. International Journal of Basic Medical Sciences and Pharmacy. 2018; 7(2), 43-57. (Line66)

21. Li D C, Papageorgiou A C. Cellulases from thermophilic fungi: recent insights and biotechnological potential. Enzyme Research. 2019; 395-417. (Line72)

22. Xu X, Fan C, Song L, Li J, Chen Y, Zhang Y, et al. A novel CreA-mediated regulation mechanism of cellulase expression in the thermophilic fungus Humicola insolens. International Journal of Molecular Sciences. 2019; 20(15): 3693. (Line72)

30. Fang H, Zhao R, Li CF, Zhao C. Simultaneous enhancement of the beta–exo synergism and exo–exo synergism in Trichoderma reesei cellulase to increase the cellulose degrading capability. Microbial Cell Factories. 2019; 18(1): 9-23. (Line103)

Response to Reviewer 2:

Reviewer #2: I recommend this manuscript for publication in PLOS ONE. I suggest the following issue to address though. Speaking about the synergistic action, the Authors should make some thoughts and provide chemical scheme of hydrolysis reaction pathway for this scenario. It will simplify understanding the concept of enzymes action.

Answer: We would like to take this opportunity to thank Reviewer 2 for his/her suggestions on my paper with kindly words. And the chemical scheme of hydrolysis reaction pathway of synergistic action was as follows: 

“Cooperation between many functionally related proteins is very important in biology. A prominent example is the apparent rate of substrate conversion increased by the addition of different combination cellulase enzymes to break down cellulose. A high level of coordination between the enzyme, e.g. exo/endo, exo/exo and endo/the BG synergistic effect, which is effective to hydrolyze crystalline cellulose required. [1] [2] [3] [4]

There are three significant types of cellulases to be used basically for the efficient degradation of cellulose. And perhaps the most widely known mechanism remains the so-called endo/exo synergy model. In this form, cellobiohydrolases (CBHs), also called exo-glucanases, cleave cellobiose from the reducing and nonreducing ends producing crystalline ends of cellulose; Endo-glucanases (EGs) break recurrently the glycosidic bonds inside the amorphous part of the substrate creating new attack sites for the CBHs. The products of cellobiohydrolases and endoglucanases are able to inhibit the activities of their own enzymes, so finally, beta-glucosidases (BGLs) hydrolyze unconfined cellobiose and various soluble cellodextrins into glucose monomers. The efficient hydrolysis of cellulose requires the composition of a complete and balanced cellulase, with all components of the cellulase co-acting the enzymatic hydrolysis of cellulose. Synergy is used to describe the ability of collaboration between enzymes. [4] [5] [6] [7] [8]

For my experiments, the enzymes of TlCel5A and TlCel6A were combined at different ratios, and the commercial β-glucosidase from Aspergillus fumigatu (Sigma-Aldrich) was added at the concentration of 10% (mol β-glucosidase/mol total enzyme) to hydrolyze cellobiose to glucose finally. When using Avicel, phosphoric acid swollen cellulose (PASC) or steam-exploded corn straw (SECS) as the substrate, combination of TlCel5A and TlCel6A showed significant synergistic action, releasing more reduced sugars than the individual enzymes. (Line270)

1. Shang B Z, Chu J W. Kinetic modeling at single-molecule resolution elucidates the mechanisms of cellulase synergy. ACS Catalysis. 2014; 4(7): 2216-2225.

2. Van Dyk J S, Pletschke B I. A review of lignocellulose bioconversion using enzymatic hydrolysis and synergistic cooperation between enzymes-factors affecting enzymes, conversion and synergy. Biotechnology Advances. 2012; 30(6):1458-1480.

3. Yang B, Dai Z, Shi-You Ding, et al. Enzymatic hydrolysis of cellulosic biomass. Biofuels. 2011; 2(4):421-450.

4. Kamal S, Khan S U, Muhammad N, et al. Insights on heterologous expression of fungal cellulases in Pichia pastoris. Biochemistry and Molecular Biology. 2018; 3(1): 1-15.

5. Olsen J P, Borch K, Westh P. Endo/exo-synergism of cellulases increases with substrate conversion. Biotechnology and bioengineering. 2017; 114(3): 696-700.

6. Badino S F, Christensen S J, Kari J, et al. Exo-exo synergy between Cel6A and Cel7A from Hypocrea jecorina: Role of carbohydrate binding module and the endo-lytic character of the enzymes. Biotechnology and bioengineering. 2017; 114(8): 1639-1647.

7. Boisset C, Fraschini C, Schülein M, et al. Imaging the enzymatic digestion of bacterial cellulose ribbons reveals the endo character of the cellobiohydrolase Cel6A from Humicola insolens and its mode of synergy with cellobiohydrolase Cel7A. Applied & Environmental Microbiology. 2000; 66(4): 1444-1452.

8. Fang H, Zhao R, Li C, et al. Simultaneous enhancement of the beta-exo synergism and exo-exo synergism in Trichoderma reesei cellulase to increase the cellulose degrading capability. Microbial cell factories. 2019; 18(1): 1-9.

---

## [Editor Report · Decision Letter 1]

23 Oct 2019

Characterization of two thermophilic cellulases from Talaromyces leycettanus JCM12802 and their synergistic action on cellulose hydrolysis

PONE-D-19-21437R1

Dear Dr. Luo,

We are pleased to inform you that your manuscript has been judged scientifically suitable for publication and will be formally accepted for publication once it complies with all outstanding technical requirements.

With kind regards,

Jean-Guy Berrin

Academic Editor

PLOS ONE
---

## [Editor Report · Acceptance letter]

1 Nov 2019

PONE-D-19-21437R1 

Characterization of two thermophilic cellulases from Talaromyces leycettanus JCM12802 and their synergistic action on cellulose hydrolysis 

Dear Dr. Luo:

I am pleased to inform you that your manuscript has been deemed suitable for publication in PLOS ONE. Congratulations! Your manuscript is now with our production department. 

With kind regards,

on behalf of

Dr Jean-Guy Berrin 

Academic Editor

PLOS ONE